# Kinetic Modeling of Glycerol Hydrogenolysis: A Short Review

**Yangzi Liu [1], Jiayu Liu [1], Zhihao Xing [1], Xueqian Zhang [1], Chen Luo [2],*, Wenjuan Yan [1],* and Xin Jin [1],***

[1] State Key Laboratory of Heavy Oil Processing, College of Chemical Engineering, China University of Petroleum, No. 66 Changjiang West Road, Qingdao 266580, China

[2] Petrochemical Research Institute, PetroChina, Beijing 102200, China

* Correspondence: luochen@petrochina.com.cn (C.L.); wenjuanyan@upc.edu.cn (W.Y.); jamesjinxin@upc.edu.cn (X.J.)

**Abstract:** Glycerol hydrogenolysis represents one of the most promising technologies for future bio-refineries. In this context, kinetic modeling provides key quantitative assessment of the significance of various reactions for process development. However, as of present, there are only limited studies on detailed kinetic modeling of glycerol conversion to 1,2-propanediol, ethylene glycol and other alcoholic products. In this work, a comprehensive summary on kinetic modeling of glycerol hydrogenolysis has been conducted to reveal the possible mechanism involved in the activation of the C-H and C-O bond in glycerol molecules. In particular, power law and Langmuir–Hinshelwood model types have been critically discussed with mechanistic insights. The outcome of this review article will offer alternative views on the scale-up design of glycerol hydrogenolysis to glycols, as well as hydrogenolysis of various other bio-derived compounds to value-added chemicals.

**Keywords:** hydrogenolysis; glycerol; kinetics





## 1. Introduction

### 1.1. Glycerol Hydrogenolysis: Key Technology for Downstream Renewables

As one of the most popular platform compounds, glycerol has been extensively studied as a model molecule for synthesis of a variety of different value-added fuels and chemicals [1,2]. Glycerol is known to be derived as an important by-product during the production of first generation bio-diesel [3]. The key chemical process, namely transesterification, yields almost 10 wt% glycerol as a co-product and formulation of methyl or ethyl fatty acids as the main component for bio-diesel molecules. Glycerol molecules consist of three carbons and three hydroxyl groups, which are considered as a most simple but well-representing compound for cellulosic biomass feedstocks [4]. Hydrogenolysis, dehydration, oxidation, as well as the reformation of glycerol can produce a variety of valuable chemical products [5]. Among all possible downstream derivatives, 1,2-propanediol (P), 1,3-propanediol (1,3-PDO) and ethylene glycol (EG), which are often derived from hydrogenolysis processes, have received the most attention from both academia and industry (Figure 1).

Hydrogenolysis of glycerol represents a classic way for transforming polyols into renewable chemicals. In this context, hydrogenolysis of glycerol mainly produces P, 1,3-PDO and EG as key polymer monomers for the plastics industry. The following paragraphs will discuss the critical role of each of the above-mentioned intermediates in the chemical industry.

**P**: P is a promising chemical with numerous applications including its use as an antifreeze, cosmetic agent, moisturizer, solvent, surfactant, and a preservative [6]. It has been conventionally manufactured through the hydrolysis of propylene oxide or co-production of dimethyl carbonate through $CO_2$ route. With a global production capacity exceeding 5 million tons annually, P is still heavily dependent on fossil fuel conversion routes. However, multiple energy intensive steps had motivated early researchers to seek

alternative ways from renewable feedstocks. Owing to similar chemical structure, glycerol hydrogenolysis to P has been extensively studied over numerous supported metal catalysts, including Ru [5,7], Pt [8], Ni [9,10], Cu [11–13] and Co [14], with solid supports ranging from carbon-based materials to metal oxides with nanostructures. In the past decades, liquid phase hydrogenolysis of glycerol to P has been studied extensively. Multinational companies such as Archer Daniel Midland have ventured into the production of P with 0.1 million tons per annum by the liquid phase hydrogenolysis of glycerol [15].

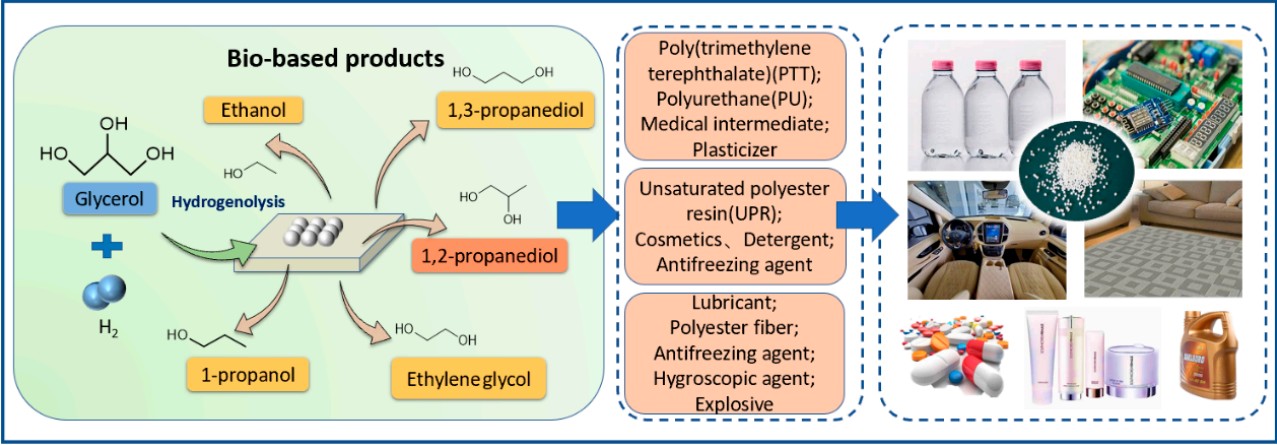

**Figure 1.** Hydrogenolysis products of glycerol.

**1,3-PDO**: In comparison to P, 1,3-PDO molecules have two hydroxyl groups at two terminal carbons. It is the main component for PTT products, which are widely used for carpets and membranes in everyday use. DuPont patented the technology for the manufacture of 1,3-PDO from glucose in early 1970s. In the past decades, despite worldwide efforts to develop glycerol-based chemistry replacements, DuPont's technology, the most advanced biological technique to our best knowledge, is still in the pilot plant stage with several engineering bottlenecking issues for stable and cost-effective production of 1,3-PDO. PChemical synthesis of 1,3-PDO is still at laboratory scale worldwide [16,17].

**EG**: EG is traditionally produced via hydrolysis of ethylene epoxide in the petrochemical industry, or hydrogenation of oxalate during coal conversion. EG has a global production capacity probably exceeding 10 million tons annually [18,19]. The limited application areas in PET plastics and restricted standard for product purity for polymerization has hindered application of EG in the chemical industry. However, bio-EG can be further reacted to produce bioethanol. The economic value of bio-ethylene produced from bioethanol dehydration is significant due to its extensive usage in the petrochemical industry [20]. EG is often formulated as a co-product during glycerol hydrogenolysis.

### 1.2. Conversion Routes of Glycerol Hydrogenolysis

Glycerol hydrogenolysis involves complicated parallel and consecutive reaction pathways producing a variety of different products. As shown in Figure 2, glycerol conversion can be initiated by a dehydrogenation reaction on the surface of metal catalysts (e.g., Ru, Cu, Ni), generating glyceraldehyde as the key intermediate. Glyceraldehyde is dehydrated to 2-hydroxyacrolein and then hydrogenated to form acetol.

Glycerol can also undergo a dehydration reaction to form acetol over acidic catalysts ($Al_2O_3$, zeolite, etc.). Acetol is then easily hydrogenated into P as the next product. Similar to the case with EG, further hydrogenolysis of P yields 1-propanol and 2-propanol as products.

EG can be obtained by the cracking of C-C during the hydrogenolysis of glycerol. It is important to mention that further hydrogenolysis of EG could provide ethanol as the final product, while methanol is the precursor for methane over noble metal catalysts.

**Figure 2.** Comprehensive reaction network for glycerol hydrogenolysis.

Literature reviews have also revealed that, RhRe-, IrRe- and Pt-WO$_x$-based catalysts can selectively facilitate cleavage of C-O at the middle carbon position, thus 1,3-PDO can be formulated as one of the main products [21,22]. However, hydrogenolysis involving C-O bond rupture of terminal carbon competes with this route.

### 1.3. Kinetic Modeling: Opportunities and Challenges

Kinetic modeling remains the core technique in the discipline of chemical reaction engineering, linking the bench-scale experimental data to pilot plant operation and scale-up design. Despite decade-long investigations on various solid catalyst materials, the progress on kinetic modeling is very limited. Table 1 summarizes the categories of supported metallic catalysts for glycerol conversion in batch and continuous reactors. It is seen that, although a total of eight types of metallic catalytic systems have been developed for glycerol hydrogenolysis, kinetic modeling is primarily focused on Cu and Ru systems.

**Table 1.** Categories of supported metallic catalysts for glycerol conversion in batch and continuous reactors.

| Cat. (Supporter) | Promoter | Kinetic | Batch | Continuous | Ref. |
|---|---|---|---|---|---|
| Cu (Al$_2$O$_3$, Cr$_2$O$_3$, MgO, SiO$_2$, ZnO, ZrO$_2$) | Ni, Pd | √ | √ | √ | [23–34] |
| Co (C, ZnO, Al$_2$O$_3$) | Re, Pd | √ | √ |  | [14,35–37] |
| Ni (C, Al$_2$O$_3$) | Cu, Ce, Ag | √ | √ | √ | [3,38–40] |
| Ru (C, Al$_2$O$_3$, TiO$_2$) | Re, Co | √ | √ | √ | [5,41–46] |
| Pt (C, Al$_2$O$_3$) | - |  | √ |  | [8,9,46] |
| Pd (Cr$_2$O$_3$, ZrO$_2$, ZnO) | Cu, Re, Co | √ | √ | √ | [36,47,48] |
| Rh (Al$_2$O$_3$, SiO$_2$) | - |  | √ |  | [49] |
| Ir (Al$_2$O$_3$) | - |  | √ |  | [50] |

To our best knowledge, no relevant discussion has been published to summarize recent developments in kinetic modeling of glycerol hydrogenolysis. Therefore, in this work, the following aspects will focus on critical discussion of kinetic modeling:Pl

(a) Model development on power law (PL) and mechanistic models for glycerol conversion; (b) metal-dependent kinetic behaviors for C-O and C-H cleavage over various metallic catalysts; (c) advances of kinetic modeling describing catalyst performances in continuous reactors.

## 2. Kinetic Modeling on Cu-Based Catalysts

Numerous works have been focused on developing mechanistic models over Cu-based catalysts. In particular, $Cu_{0.45}Zn_{0.15}Mg_{5.4}Al_2O_9$, Cu/MgO and Cu/SiO$_2$ catalysts have been investigated for kinetic modeling. Power law models have been derived and validated on those three types of Cu catalysts (Table 2). It was found that, the reaction order of glycerol is 1st, 1.2th and 0.27th over Cu/ZnO-MgO-Al$_2$O$_3$, Cu/MgO and Cu/SiO$_2$ catalysts, respectively.

**Table 2.** Power law models for glycerol hydrogenolysis over Cu-based catalysts.

| No. | Catalyst | Condition | X (%) | S$_P$ (%) | Kinetic Model | | Ref. |
|---|---|---|---|---|---|---|---|
| | | | | | E$_a$ (kJ·mol$^{-1}$) | Expression | |
| 1 | $Cu_{0.45}Zn_{0.15}Mg_{5.4}Al_2O_9$ | Temperature: 190–250 °C<br>H$_2$ pressure: 3.5–5 MPa<br>Glycerol concentration: 20 wt%<br>Catalyst loading: 5.4 wt%<br>Time: 12 h | 100 | 93.7 | G-P: 35.1 | $r_G = -\frac{dC_G}{dt} = k_G \exp\left[\frac{-E_a}{RT}\right]C_G$ | [30] |
| 2 | Cu/MgO | Temperature: 190–230 °C<br>H$_2$ pressure: 3–6 MPa<br>Glycerol concentration: 20–60 wt%<br>Catalyst loading: 35 wt%<br>Time: 2–12 h | 96 | 89 | G-P: 84.9 | $r_G = -\frac{dC_G}{dt} = k_G \exp\left[\frac{-E_a}{RT}\right]C_G^{1.2}$ | [27] |
| 3 | Cu/SiO$_2$ | Temperature: 180–240 °C<br>H$_2$ pressure: 2–8 MPa<br>Glycerol concentration: 28–50 wt%<br>Catalyst loading: 18 wt% | - | 95 | G-P: 94.3 | $r_G = k_G \exp\left[\frac{-E_a}{RT}\right]C_G^{0.27}C_{H_2}^{0.95}$<br>$r_{G-P} = k_{G-P} \exp\left[\frac{-E_a}{RT}\right]C_G^{0.17}C_{H_2}^{1.06}$ | [31] |

Mechanistic models involving the adsorption of glycerol, molecular H$_2$, P and EG species were considered for Langmuir-Hinshelwood-Hougen-Watson (LHHW) behaviors (Table 3). A total of six different mechanistic models have been derived and statistically fitted with experimental data over $Cu_{0.45}Zn_{0.15}Mg_{5.4}Al_2O_9$, Cu/MgO, Cu/SiO$_2$, Cu/ZnO-ZrO$_2$-Cr$_2$O$_3$ and Cu/ZnO-Al$_2$O$_3$ catalysts (Table 4). In particular, Pandhare and colleagues proposed a dual-site mechanism over Cu/MgO catalysts, considering adsorption of glycerol and molecular H$_2$ species [27]. Sharma and co-workers proposed and validated another mechanism involving the adsorption of glycerol, molecular H$_2$, P and water on catalyst surface over Cu/ZnO-ZrO$_2$-Cr$_2$O$_3$ catalysts [28]. The activation barrier is approximately 131.9 kJ·mol$^{-1}$. However, Zhou and colleagues believed that adsorption of glycerol, acetol and P occurs on a different type of site compared to molecular H$_2$ species [29].

This part of the summary will critically discuss the derivation and validation of various types of power law and LHHW models over Cu catalysts. It should be noted that many studies have shown that the valence of non-noble metal components in the catalyst will affect the hydrogenolysis performance of the catalyst. However, the hydrogen condition may affect the reduction degree of non-precious metals to a certain extent, which will complicate the hydrogen pressure kinetics [51]. Unfortunately, this part of the work has yet to be established systematically, so it could be a good topic for future work.

### 2.1. Power Law Models

A bi-functional layered double hydroxide (LDH) catalyst $Cu_{0.45}Zn_{0.15}Mg_{5.4}Al_2O_9$ was synthesized by the urea hydrolysis method, reported by Meena et al. [30]. The effect of the reaction parameters such as temperature (190–250 °C), H$_2$ pressure (3.5–5 MPa), and catalyst weight on conversion and selectivity were determined in a batch reactor. The results show that with the temperature increasing, conversion of glycerol was found to be increased, while the selectivity of P increased first ($\leq$210 °C) and then decreased ($\geq$220 °C) due to other hydrogenolysis reactions. The glycerol conversion rate increased with pressure because there were more H$_2$ molecules available around the glycerol. Higher pressure (>4.5 MPa) is favorable for the formation of degradation products of P, resulting in decreased selectivity of P.

**Table 3.** Mechanistic models for glycerol conversion over Cu-based catalysts.

| No. | Catalyst | Condition | X (%) | $S_P$ (%) | Kinetic Model | | Ref. |
|---|---|---|---|---|---|---|---|
| | | | | | $E_a$ (kJ·mol$^{-1}$) | Expression | |
| 1 | $Cu_{0.45}Zn_{0.15}Mg_{5.4}Al_2O_9$ | Temperature: 210 °C<br>$H_2$ pressure: 3.5–5 MPa<br>Glycerol concentration: 20 wt%<br>Catalyst loading: 8 wt%<br>Time: 12 h | 100 | 93.7 | G-P: 35.1 | $r_{G-P} = \dfrac{k_{G-P} K_{H_2} P_{H_2}}{16 K_G^3 C_G^3}$ <br> $\left[\left(1 + \left(P_{H_2} K_{H_2}\right)^{0.5} + K_P C_P\right)^2 + \left(4 K_G C_{T\$} C_G\right)^{0.5} - \left(1 + \left(P_{H_2} K_{H_2}\right)^{0.5} + K_P C_P\right)\right]^4$ | [30] |
| 2 | Cu/MgO | Temperature: 190–230 °C<br>$H_2$ pressure: 3–6 MPa<br>Glycerol concentration: 20–60 wt%<br>Catalyst loading: 35 wt%<br>Time: 2–12 h | 96 | 89 | G-P: 84.9 | $r_{G-P} = \dfrac{k_{G-P} C_G P_H}{\left(1 + K_G C_G + K_H P_H + K_P C_P + K_{EG} C_{EG}\right)^2}$ | [27] |
| 3 | $Cu/ZnO\text{-}ZrO_2\text{-}Cr_2O_3$ | Temperature: 220–250 °C<br>$H_2$ pressure: 1–4 MPa<br>Glycerol concentration: 60–100 wt%<br>Catalyst loading: 3 wt%<br>Time: 10 h | 100 | 97 | G-P: 131.9 | $r_{G-P} = \dfrac{k_{G-P} K_G K_{H_2} C_{T\$}^2 C_G P_{H_2}}{\left(1 + K_G C_G + K_{H_2} P_{H_2} + K_P C_P + K_{EG} C_{EG}\right)^2}$ | [28] |
| 4 | $Cu/ZnO\text{-}Al_2O_3$ | Temperature: 220–240 °C<br>$H_2$ pressure: 3–5 MPa<br>Glycerol concentration: 60–100 wt%<br>Catalyst loading: 37 wt% | 81.5 | 93.4 | G-A: 86.6 | $r_{G-A} = \dfrac{k_{G-A} K_G C_G}{1 + K_G C_G + C_A K_A + K_P C_P}$ | [29] |
| | | | | | A-P: 57.8 | $r_{A-P} = \dfrac{k_{A-P} K_A C_A K_{H_2} P_{H_2}}{\left(1 + K_G C_G + C_A K_A + K_P C_P\right)\left(1 + \left(K_{H_2} P_{H_2}\right)^{0.5}\right)^2}$ | |
| 5 | $Cu/ZnO\text{-}Al_2O_3$ | Temperature: 200–270 °C<br>$N_2$ pressure: 3 MPa<br>Glycerol concentration: 1–5 wt%<br>Catalyst loading: 49 wt%<br>Time: 0–1.25 h | 95.6 | 79.4 | G-A: 87 | $r_{G-A} = \dfrac{k_{G-A} C_{T\$}^2 K_G C_G}{\left[1 + K_G C_G + K_{CH_3OH} C_{CH_3OH} + K_A C_A + K_P C_P + K_{H_2} C_{H_2} + K_{AH} C_{AH} + K_W C_W + K_{EG} C_{EG} + \frac{K_{OH} K_W C_W}{\left(K_{H_2} C_{H_2}\right)^{0.5}}\right]^2}$ | [32] |
| | | | | | A-P: 68.4 | $r_{A-P} = \dfrac{k_{A-P} C_{T\$}^2 \left(K_{AH} K_A C_A K_{H_2} C_{H_2} - K_{A-P}^{-1} K_P C_P\right)}{\left[1 + K_G C_G + K_{CH_3OH} C_{CH_3OH} + K_A C_A + K_P C_P + K_{H_2} C_{H_2} + K_{AH} C_{AH} + K_W C_W + K_{EG} C_{EG} + \frac{K_{OH} K_W C_W}{\left(K_{H_2} C_{H_2}\right)^{0.5}}\right]^2}$ | |
| 6 | $Cu/ZnO\text{-}Al_2O_3$ | Temperature: 200–270 °C<br>$N_2$ pressure: 10 MPa<br>Glycerol concentration: 1.3–5 wt%<br>Catalyst loading: 49 wt%<br>Time: 0–1.25 h | 95.6 | 79.4 | G-A: 87 | $r_{G-A} = \dfrac{k_{G-A} K_{AH} K_G C_G}{\left[1 + K_G C_G + K_{CH_3OH} C_{CH_3OH} + K_A C_A + K_P C_P + \left(K_{H_2} C_{H_2}\right)^{0.5} + K_{AH} K_A C_A \left(K_{H_2} C_{H_2}\right)^{0.5}\right]^2}$ | [33] |
| | | | | | A-P: 68.4 | $r_{A-P} = \dfrac{k_{A-P} \left(K_{AH} K_A C_A K_{H_2} C_{H_2} - K_{A-P}^{-1} K_P C_P\right)}{\left[1 + K_G C_G + K_{CH_3OH} C_{CH_3OH} + K_A C_A + K_P C_P + \left(K_{H_2} C_{H_2}\right)^{0.5} + K_{AH} K_A C_A \left(K_{H_2} C_{H_2}\right)^{0.5}\right]^2}$ | |

**Table 3.** *Cont.*

| No. | Catalyst | Condition | X (%) | $S_P$ (%) | Kinetic Model | | Ref. |
|---|---|---|---|---|---|---|---|
| | | | | | $E_a$ (kJ·mol$^{-1}$) | Expression | |
| 7 | Cu-based | Temperature: 190–240 °C<br>H$_2$ pressure: 6.5–8 MPa<br>Glycerol concentration: 99.5 wt%<br>Space times (W/FG$^0$):<br>25–340 kg·s·mol$^{-1}$ | 75 | 90 | G-A: 84 | $r_{G-A} = C_{T\$}^2 k_{G-A}\left(\theta_* \theta_{G*} - K_{G-A}{}^{-1}\theta_{A*}\theta_{H_2O*}\right)$ | [34] |
| | | | | | A-P: 59 | $r_{A-P} = C_{T\$}^2 k_{A-P}\left(\theta_{AH*}\theta_{H*} - K_{A-P}{}^{-1}\theta_{P*}\theta_*\right)$ | |

**Table 4.** Mechanistic description for glycerol conversion models.

| Dual-Site Mechanism [30] | | | Single-Site Mechanism for P and EG Formation [27] | | | With Acetol as the Intermediate [34] | | |
|---|---|---|---|---|---|---|---|---|
| 1st step.<br>Adsorption on site | $G + 2\theta \leftrightarrow \theta\cdot G\cdot\theta$<br>$H_2 + 2\theta \leftrightarrow 2H\cdot\theta$ | $K_G$<br>$K_{H2}$ | 1st step.<br>Adsorption on site | $G + \theta \leftrightarrow G\cdot\theta$<br>$H + \theta \leftrightarrow H\cdot\theta$ | $K_G$<br>$K_H$ | 1st step.<br>Adsorption on site | $G + \theta \leftrightarrow G\cdot\theta$<br>$H_2 + 2\theta \leftrightarrow 2H\cdot\theta$ | $K_G$<br>$K_{H2}$ |
| 2nd step.<br>Surface reaction | $\theta\cdot G\cdot\theta + 2H\cdot\theta \leftrightarrow P\cdot\theta + 3\theta + W$ | $r_{G-P}, k_{G-P}$ | 2nd step.<br>Surface reaction | $G\cdot\theta + H\cdot\theta \leftrightarrow P\cdot\theta + W\cdot\theta$<br>$G\cdot\theta + H\cdot\theta \leftrightarrow E\cdot\theta + \theta$ | $r_{G-P}, k_{G-P}$<br>$r_{G-EG}, k_{G-EG}$ | 2nd step.<br>Surface reaction | $G\cdot\theta + \theta \leftrightarrow A\cdot\theta + W\cdot\theta$<br>$A\cdot\theta + H\cdot\theta \leftrightarrow AH\cdot\theta + \theta$<br>$AH\cdot\theta + H\cdot\theta \leftrightarrow P\cdot\theta + \theta$ | $r_{G-A}, k_{G-A}$<br>$K_{AH}$<br>$r_{A-P}, k_{A-P}$ |
| 3rd step.<br>Desorption | $P\cdot\theta \leftrightarrow P + \theta$ | $K_P$ | 3rd step.<br>Desorption | $P\cdot\theta \leftrightarrow p + \theta$<br>$E\cdot\theta \leftrightarrow E + \theta$ | $K_P$<br>$K_{EG}$ | 3rd step.<br>Desorption | $A\cdot\theta \leftrightarrow A + \theta$<br>$P\cdot\theta \leftrightarrow p + \theta$<br>$W\cdot\theta \leftrightarrow W + \theta$ | $K_A$<br>$K_P$ |
| **Dual-Site Mechanism [29]** | | | **Single-Site Mechanism Considering Water Adsorption [28]** | | | **APR of Methanol for In-Situ H$_2$ [32,33]** | | |
| 1st step.<br>Adsorption on site | $G + \theta \leftrightarrow G\cdot\theta$<br>$H_2 + 2\$ \leftrightarrow 2H\cdot\$$ | $K_G$<br>$K_{H2}$ | 1st step.<br>Adsorption on site | $G + \theta \leftrightarrow G\cdot\theta$<br>$H_2 + 2\theta \leftrightarrow 2H\cdot\theta$ | $K_G$<br>$K_{H2}$ | 1st step.<br>H$_2$ formation | $W + \theta \leftrightarrow W\cdot\theta$<br>$W\cdot\theta + \theta \leftrightarrow OH^-\cdot\theta + H\cdot\theta$<br>$CH_3OH + \theta \leftrightarrow CH_3OH\cdot\theta$<br>$CH_3OH\cdot\theta + \theta \rightarrow CH_3O\cdot\theta + H\cdot\theta$<br>$CH_3O\cdot\theta + \theta \rightarrow CH_2O\cdot\theta + H\cdot\theta$<br>$CO\cdot\theta + OH^-\cdot\theta \rightarrow H\cdot\theta + CO_2$<br>$2H\cdot\theta \leftrightarrow H_2 + 2\theta$ | $K_W$<br>$K_{OH}$<br>$K_{CH3OH}$<br>$r_{CH3OH}, k_{CH3OH}$<br>$K_{H2}$ |
| 2nd step.<br>Surface reaction | $G\cdot\theta \rightarrow A\cdot\theta + H_2O$<br>$2H\cdot\$ + A\cdot\theta \rightarrow P\cdot\theta + 2\$$ | $r_{G-A}, k_{G-A}$<br>$r_{A-P}, k_{A-P}$ | 2nd step.<br>Surface reaction | $G\cdot\theta + H\cdot\theta \leftrightarrow P\cdot\theta + W\cdot\theta$ | $r_{G-P}, k_{G-P}$ | 2nd step.<br>acetol formation | $G + \theta \leftrightarrow G\cdot\theta$<br>$G\cdot\theta + \theta \rightarrow A\cdot\theta + W\cdot\theta$<br>$A\cdot\theta \leftrightarrow A + \theta$ | $K_G$<br>$r_{G-A}, k_{G-A}$<br>$K_A$ |
| 3rd step.<br>Desorption | $A\cdot\theta \leftrightarrow A + \theta$<br>$P\cdot\theta \leftrightarrow P + \theta$ | $K_A$<br>$K_P$ | 3rd step.<br>Desorption | $P\cdot\theta \leftrightarrow P + \theta$<br>$W\cdot\theta \leftrightarrow W + \theta$ | $K_P$<br>$K_W$ | 3rd step.<br>1,2-propanediol formation | $A\cdot\theta + H\cdot\theta \leftrightarrow AH\cdot\theta + \theta$<br>$AH\cdot\theta + H\cdot\theta \rightarrow P\cdot\theta + \theta$<br>$P\cdot\theta \leftrightarrow P + \theta$ | $K_{AH}$<br>$r_{A-P}, k_{A-P}$<br>$K_P$ |
| | | | | | | 4th step. ethylene glycol formation | $G\cdot\theta + H\cdot\theta \rightarrow G\cdot H + \theta$<br>$GH\cdot\theta + H\cdot\theta \rightarrow E\cdot\theta + CH_3OH\cdot\theta$<br>$E\cdot\theta \leftrightarrow E + \theta$ | $r_{GH}, k_{GH}$<br>$K_{EG}$ |

The power law model calculated data were compared with previously reported values over various catalysts (Table 2). It was observed that the pre-exponential factor ($2.35 \times 10^3$ L·gcat$^{-1}$·h$^{-1}$) and activation energy (35.1 kJ·mol$^{-1}$) value obtained in this study over the Cu$_{0.45}$Zn$_{0.15}$Mg$_{5.4}$Al$_2$O$_9$ catalyst was low.

Pandhare et al. studied the kinetics of liquid phase hydrogenolysis of glycerol to P by using 35 wt% Cu/MgO in a slurry batch reactor [27]. The catalysts were prepared by the precipitation-deposition method. They evaluated the effect of temperature (190–230 °C), pressure (3–6 MPa), and glycerol concentration (20–60 wt%) on conversion and selectivity of various products. On the basis of the reaction products obtained, two parallel routes for the formations of P and EG from glycerol were proposed and discussed.

The kinetic analysis demonstrated significant variation in glycerol conversion and product selectivity under different reaction conditions. It is observed that a low reaction temperature (<210 °C) and a short period of reaction time were beneficial for higher P selectivity (>87%). In addition, the reaction rate increased with H$_2$ pressure, while the selectivity to P and EG was not significantly affected by H$_2$ pressure. A higher glycerol concentration and a longer period of reaction time were not beneficial for improved selectivity to P.

The power law model showed that the apparent reaction order "n" for hydrogenolysis was 1.2 with respect to glycerol. The calculated activation energy and pre-exponential factor were 84.9 kJ·mol$^{-1}$ and $45.2 \times 10^7$ mol·gcat$^{-1}$·h$^{-1}$, respectively.

The solvent effect on kinetic behaviors was studied by Vasiliadou and colleagues [31]. Two parallel reaction pathways were considered to produce P and 1,3-PDO, in which P was the main product with a selectivity of 95%. Using 1-butanol as the solvent, the catalyst activity was enhanced by about 20% compared with pure glycerol in the feedstock. The change of H$_2$ concentration in liquid phase (glycerol/1-butanol mixture) was also studied. Using 1-butanlol as the solvent, the influence of reaction conditions on the reaction rate was studied by varying the parameters in the range of temperatures from 180 to 240 °C the H$_2$ pressure from 2–8 MPa and the glycerol initial concentration from 28–50 wt%.

The power law model for the overall consumption rate of glycerol showed that the apparent reaction order "n" for hydrogenolysis was 0.27 with respect to glycerol and 0.95 with respect to hydrogen. The low dependence of the reaction rate on glycerol indicates that the active sites of the catalyst are expected to be almost completely occupied by the adsorbed glycerol species.

The power law model can be estimated preliminarily using the reaction rate parameter. Although the power law model demonstrated a good fit with the experimental data, this approach had major limitations. It only considers the effect of the concentration of glycerol and hydrogen on the reaction rate. The various steps integrated with the heterogeneous catalytic process (adsorption-surface reaction-desorption) were not considered in this type of model. In addition, the substrate inhibition effect posed by concentrated reactants on the catalyst surface should also be considered, as it will significantly alter the observed reaction rates. Such important kinetic behaviors will also present strong impacts on reactor designs for process development.

### 2.2. LHHW Models

As has already been mentioned, a total of six different mechanistic models have been discussed and fitted over Cu-based catalysts (Table 4). In this part, the adsorption behaviors will be critically reviewed and compared for insights into the activation mechanism. The mechanism model of glycerol conversion on Cu-based catalysts is summarized in Table 3, with respect to catalyst type, reaction temperature, pressure, percent conversion of glycerol, selectivity of p, activation energy and reaction kinetics equation.

Meena and colleagues proposed a dual site LHHW model which considers the adsorption of glycerol on two sites, dissociative adsorption of molecular H$_2$, and P molecules [30]. The following rate equation was derived and validated for glycerol conversion to P.

$$r_{G-P} = \frac{k_{G-P}K_{H_2}P_{H_2}}{16K_G^3C_G^3}\left[\left(1 + \left(P_{H_2}K_{H_2}\right)^{0.5} + K_PC_P\right)^2 + \left(4K_GC_{T\$}C_G\right)^{0.5} - \left(1 + \left(P_{H_2}K_{H_2}\right)^{0.5} + K_PC_P\right)\right]^4$$

In another model, Pandhare and co-workers validated the mechanism involving the adsorption of glycerol, dissociative activation of molecular $H_2$ and strong interaction of P and EG with catalyst surface [27]. The following kinetic equations were validated for the formation of P and EG. It is clear that formation of P and EG occurs on different types of active sites over Cu/MgO catalysts. However, the details on the $H_2$ pressure measurement were not illustrated in this work.

$$r_{G-P} = \frac{k_{G-P}C_GP_H}{(1 + K_GC_G + K_HP_H + K_PC_P + K_{EG}C_{EG})^2}$$

$$r_{G-EG} = \frac{k_{G-EG}C_GP_H}{(1 + K_GC_G + K_HP_H + K_PC_P + K_{EG}C_{EG})^2}$$

A series of complicated kinetic models which consider tandem methanol reformation and hydrogenolysis of glycerol were also proposed and validated over Cu/ZnO-Al$_2$O$_3$ catalysts [32]. Two paths of glycerol hydrogenolysis from intermediate acetol to main product P were proposed. The elucidation of reaction pathways using an in-situ IR technique showed that under the condition of molecular $H_2$, acetol could be produced via direct dehydration by glycerol, while under the condition of depleting $H_2$ acetol would be produced via glycerol dehydrogenation-dehydration forming glyceraldehyde and 2-hydroxyacrolein as intermediates (Figure 2).

In the presence of glycerol and methanol in the feed, it was shown that the two reactants compete for the same active center and that the adsorption of glycerol on the catalyst surface was stronger. At the beginning of the reaction, the high concentration of glycerol prevented the methanol reaction from producing $H_2$, thus the glycerol was forced to dehydrogenate and then dehydrate to produce the desired $H_2$. As the reaction proceeds, the concentration of glycerol decreases, and more methanol is adsorbed on the active site of the catalyst. It is revealed that the combined reaction cycle proceeds in four steps: (1) $H_2$ production via methanol aqueous phase reforming (APR), (2) glycerol dehydration to acetol, (3) acetol hydrogenation to 1,2-propanediol and (4) ethylene glycol formation via C-C bond cleavage (Figure 3).

**Figure 3.** Simplified scheme of glycerol hydrogenolysis reaction mechanism.

The same approach as in the previous article simplifies the reaction path [33]. It was considered that all the reactions take place on a metallic $Cu^0$ surface. The hypothesis of the reaction mechanism is described in the case of the Langmuir–Hinshelwood model.

In this model, all molecules are adsorbed and activated on the surface of a Cu catalyst before participating in surface reactions. On the surface of the Cu/ZnO-Al$_2$O$_3$ catalyst, APR (aqueous-phase reforming) of methanol generates CO through continuous dehydrogenation of methanol (forming methoxy and formate species as intermediates), and CO is further transformed to CO$_2$ through a water–gas shift reaction.

The dehydrogenation of the absorbed methanol to methoxy species was defined as the rate determining step (RDS) for the H$_2$ formation (step 1). Glycerol dehydration was defined as the RDS for acetol formation (step 2). The RDS which could best describe the formation of the P (step 3) is the second reversible hydrogenolysis step of acetol. The first hydrogenolysis step of glycerol (C-C bond scission) was chosen as the RDS for EG formation (step 4).

$$r_{G-A} = \frac{k_{G-A} K_{AH} K_G C_G}{\left[1 + K_G C_G + K_{CH_3OH} C_{CH_3OH} + K_A C_A + K_P C_P + \left(K_{H_2} C_{H_2}\right)^{0.5} + K_{AH} K_A C_A \left(K_{H_2} C_{H_2}\right)^{0.5}\right]^2}$$

$$r_{A-P} = \frac{k_{A-P}\left(K_{AH} K_A C_A K_{H_2} C_{H_2} - K_{A-P}{}^{-1} K_P C_P\right)}{\left[1 + K_G C_G + K_{CH_3OH} C_{CH_3OH} + K_A C_A + K_P C_P + \left(K_{H_2} C_{H_2}\right)^{0.5} + K_{AH} K_A C_A \left(K_{H_2} C_{H_2}\right)^{0.5}\right]^2}$$

The same research group also reported an updated mechanism for hydrogenolysis of glycerol involving APR of methanol [33]. In this work, the kinetic model was investigated under reaction conditions of 0–1.25 h and 200–270 °C, with glycerol concentrations ranging from 1–5 wt% and methanol concentrations of 7–30 wt%. Higher temperature (270 °C) is favorable for glycerol hydrogenolysis and H$_2$ production from methanol, while EG formation showed a weak temperature dependency. When the glycerol concentration increases up to 5 wt%, with the glycerol concentration increasing, the rate of formaldehyde consumption and H$_2$ production both decrease, which was due to the stronger adsorption of glycerol on the active sites than methanol.

Zhou and co-workers discussed another mechanism considering activation of glycerol and species at different types of active sites [29]. First, a series of Cu-ZnO-Al$_2$O$_3$ catalysts with different metal compositions are prepared using the co-precipitation method. The activity of the catalysts was tested in a tubular fixed bed reactor under the reaction conditions of 220–240 °C and 3–5 MPa H$_2$. The result showed that metal composition had a significant influence on the performance of catalyst. A two-step mechanism was considered to be a good description of the reaction pathway, in which glycerol is firstly dehydrated to acetol, then acetol is hydrogenated to P. As a result, a two-site Langmuir-Hinshelwood kinetic model was established.

$$r_{A-P} = \frac{k_{A-P} K_A C_A K_{H_2} P_{H_2}}{(1 + K_G C_G + C_A K_A + K_P C_P)\left(1 + \left(K_{H_2} P_{H_2}\right)^{0.5}\right)^2}$$

The reaction rate derived by the LHHW model includes all the adsorption, desorption, and surface reaction steps. Therefore, the values obtained from the LHHW model are more realistic.

### 2.3. Horiuti/Temkin Model

Thybaut and colleagues studied another type of model for glycerol hydrogenolysis [34]. In this model, a total site balance equation involves the adsorption of H$_2$, glycerol, acetol and P. In comparison to other models, this study assumed that the second conversion step for acetol is the rate limiting step for the formation of P. In further details, in the case of acetol formation, a scission of terminal C-O bond is required, thus a lower mobility of the transition state is required to enable this reaction.

$$r_{G-A} = C_{T\$}^2 k_{G-A} \left(\theta_* \theta_{G*} - K_{G-A}{}^{-1} \theta_{A*} \theta_{H_2O*}\right)$$

$$r_{A-P} = C_{T\$}^2 k_{A-P}\left(\theta_{AH*}\theta_{H*} - K_{A-P}^{-1}\theta_{P*}\theta_*\right)$$

Evidently, considering the strong adsorption of reactants and products can accurately reflect the intrinsic kinetic behaviors under various reaction conditions. The coverage of reactants and products on the catalyst surface as well as the strength (enthalpy) plays a key role in determining the surface reaction rates. It is important to mention in this part, the activation mode of molecular $H_2$ also contributes to multiphase kinetics for glycerol conversion.

However, limitations for existing LHHW types of models are also obvious. For example, the types of surface sites are still ambiguous at a molecular or atomic level. In other words, the surface adsorption sites are still defined in term of mathematical interpretation rather than chemical structures. Considering the case with Cu catalysts, as they are quickly evolving under a reductive environment, the well-defined adsorptive sites are important in order to understand the intrinsic behaviors. Therefore, combinatory studies on surface characterization and quantitative assessment will be the focus for future studies on Cu-based catalysts for glycerol conversion.

Furthermore, advances in nanoscience and nanotechnology will be helpful for chemical engineers to develop well-defined Cu-based catalysts to achieve precise evaluation on kinetic rates for C-O and C=O bond activation.

### 3. Kinetic Modeling on Ru-Based Catalysts

Compared with Cu-based catalysts, very few studies have focused on glycerol hydrogenolysis using Ru-based solid catalysts. Ru catalysts are known to be superior in C-O cleavage reactions during conversion of glycerol, xylitol, sorbitol, and 5-hydroxyl methyl furan. However, methanation is a major issue plaguing Ru catalysts for improved chemiselectivity towards P and glycols. More importantly, compared with Cu-based catalysts, the reaction network is much more complicated (Table 5).

Torres and colleagues conducted kinetic modeling over bimetallic RuRe/C catalysts [41]. Bimetallic RuRe/C and monometallic Ru/C catalyst were prepared by a precipitation method. The result showed that Re has a prominent effect as a promoter for the selectivity to P (18.9% to 36.6%). This may be because of the improved dispersion of Ru in the presence of Re [42]. They also found that Re has no activity toward the hydrogenolysis of glycerol by itself. There was more selectivity to EG (18.5% to 7.3%) in the liquid-phase products and methane (51.6% to 18.5%) gas-phase products by Ru/C catalyst. This may be due to the ability of Ru to promote undesired C-C cleavage to produce by-products [52]. RuRe bimetallic catalyst was used for further studies in a hydrogen pressure of 2.4–9.6 MPa and a temperature range of 220–240 °C. It was found that at higher $H_2$ partial pressures and higher temperatures, the conversion of glycerol was increased while the selectivity of P was decreased. This was attributed to higher hydrogenolysis activity of P to propanol and higher reforming activity to produce more gaseous products. Similar conversion and selectivity trends were observed with the change of catalyst concentration (8.33–66.67 kg·m$^{-3}$). This work has proposed a validated a power law model considering 1st order for glycerol and $H_2$ pressure, respectively.

$$r_{G-P} = k_{G-A}C_G\frac{\left(C_{H_2}\right)_g}{H_{H_2}} - k_{P-PA}C_P^{0.5}\frac{\left(C_{H_2}\right)_g}{H_{H_2}}$$

In another work, the influence of pH on kinetics has been systematically investigated for glycerol conversion. At different pH levels, the kinetics of the hydrogenolysis reaction of glycerol over Ru/C catalyst was studied in a batch reactor by Lahr and Shanks, for developing an improved mechanistic understanding of the conversion of the more complex higher polyhydric alcohols [43]. In the reactions, CaO and CaCO$_3$ were used to keep the pH at 11.7 and 8.0, respectively.

**Table 5.** Kinetic models over other supported metal catalysts.

| No. | Catalyst | Condition | X (%) | $S_P$ (%) | Kinetic Model | Kinetic Equation | | Ref. |
|---|---|---|---|---|---|---|---|---|
| | | | | | | $E_a$ (kJ·mol$^{-1}$) | Expression | |
| 1 | Co/ZnO | Temperature: 160–220 °C <br> H$_2$ pressure: 2–6 MPa <br> Glycerol concentration: 10–40 wt% <br> Catalyst loading: 20–70 wt%, Time: 8 h, <br> pH: 10 | 70 | 80 | PL | G-P: 31.08 | $r_G = -\frac{dC_G}{dt} = k_0 \exp\left[\frac{-E_a}{RT}\right] C_G^{0.7355} C_{H2}^{0.5697}$ | [35] |
| 2 | RuRe/C | Temperature: 200–230 °C <br> H$_2$ pressure: 2.4–9.6 MPa <br> Glycerol concentration: 10 wt% <br> Catalyst loading: 1 wt% Ru, 1 wt% Re <br> Time: 1–6 h | 57.7 | 36.6 | PL | G-P: 54.2 | $r_{G-P} = k_{G-A} C_G \frac{(C_{H_2})_g}{H_{H_2}} - k_{P-PA} C_P^{0.5} \frac{(C_{H_2})_g}{H_{H_2}}$ | [41] |
| 3 | Ru/C | H$_2$ pressure: 7 MPa <br> Glycerol concentration: 10–15 wt% <br> Catalyst loading: 5 wt% <br> pH: 11.7 (CaO) and 8.0 (CaCO$_3$) | - | 19 | LH | - | $r_{iG} = \frac{S_{iG} k_G C_G^{1.5} + k_{iG} C_G}{K_G C_G + k_{EG} C_{EG} + k_P C_P + 1}$ | [43] |
| 4 | CuNi/Al$_2$O$_3$ | Temperature: 220 °C <br> H$_2$ pressure: 0.75 MPa <br> Glycerol concentration: 20 wt% <br> Catalyst loading: 20 wt% <br> Contact times: W/F$_{Ao}$ = 101–811 <br> kgcat·h·kmol$^{-1}$ | 100 | 89.5 | Eley–Rideal | G-A: 55.14 | $r_{G-A} = \frac{k_{G-A} K_G P_G C_{TS}}{1 + K_G P_G + K_A P_A + P_P K_P}$ | [53] |
| | | | | | | A-P: 50.87 | $r_{A-P} = \frac{k_{A-P} P_A P_{H_2} C_{TS}}{K_A (1 + K_G P_G + K_A P_A + P_P K_P)}$ | |
| 5 | PdReCo/C | Temperature: 180–203 °C <br> H$_2$ pressure: 3.3–13.3 MPa <br> Glycerol concentration: 40 wt% <br> Catalyst loading: 2.5 wt% Co, 0.5 wt% <br> Pd, and 2.4 wt% Re | 96 | - | Trickle-bed model | G-P: 86 | $r_G = \frac{k_G C_G C_{OH-} C_{H_2}^2}{C_G C_{OH-} + K_{H_2} C_{H_2}^3}$ | [48] |
| 6 | Pd/m-ZrO$_2$ + ZnO | Temperature: 220 °C <br> H$_2$ pressure: 6.0 MPa <br> Catalyst loading: 1 wt% <br> Glycerol concentration:10 wt% <br> Time: 4 h | 40 | 94.1 | - | - | $r_G = \frac{k_G C_G P_{H_2}^{-0.5}}{\left(1 + K_P C_G + K_{Alk} C_G P_{H_2}^{-0.5} + K_{H_2}^{-0.5} P_{H_2}^{0.5}\right)^2}$ | [36] |



A kinetic study of EG and P degradation was carried out because under hydrogenolysis conditions, reaction products such as EG and P tend to react further to generate alcohols and alkanes. The degradation rate of diols was calculated at different pHs. The results have shown that the average reaction rates for EG and P were 11 and 14 $mol \cdot g_{cat}^{-1} \cdot h^{-1}$, respectively. The reason may be that the presence of the nonoxygenated end in P causes it to be partially repelled by the catalytic surface [54]. The degradation mechanism of EG can be described in the following steps:

For step 1, the EG molecule is adsorbed on the surface of the catalyst and dehydrogenated into aldehydes or ketones, which then pass through C-C or a C-O cleavage, known as the retro-aldol mechanism. Experimental results demonstrated that pH of the system affected the overall rate. Since no aldehydes or ketones were detected during the reaction and the initial concentration of EG did not affect the reaction rate, step 2 proved to be the controlling step of the reaction. On the contrary, since the hydrogenation step is not a limiting step of the reaction, the model does not take into account the effect of $H_2$ on the reaction, and previous reports have indicated that $H_2$ does not cover a large area of the catalyst surface in similar reactions.

Based on the above-mentioned mechanism, a Langmuir-Hinshelwood kinetic model was developed to describe the degradation of EG and P.

$$r_{iG} = \frac{k_{iG}C_{iG}}{K_{EG}C_{EG} + K_P C_P + 1}$$

The kinetics of hydrogenolysis reaction was studied. The experimental results show that adding EG can reduce the reaction rate of glycerol, so the problem of competitive adsorption should be considered. Assuming that glycerol takes a similar path as EG and P, glycerol was first dehydrogenated to an aldehyde or ketone. The reaction rate of glycerol can be expressed in the following form:

$$-r_G = \frac{k_{iG}C_G}{K_G C_G + K_{EG}C_{EG} + K_P C_p + 1}$$

The simulation results obtained by this model had a better agreement with the experimental results, but the mechanism of reaction order of 1.5 has not been explained in detail.

$$-r_G = \frac{k_G C_G^{1.5}}{K_G C_G + K_{EG}C_{EG} + K_P C_p + 1}$$

Finally, the rate equation of the decreasing solution of EG and P in glycerol was obtained.

$$r_{iG} = \frac{S_{iG}k_G C_G^{1.5} - k_{iG}C_G}{K_G C_G + k_{EG}C_{EG} + k_P C_P + 1}$$

Although the Ru catalyst shows high activity for glycerol hydrogenolysis, it also promotes unwanted C-C cleavage. This is the main cause of by-products (EG, methane and methanol) formation.

## 4. Kinetic Modeling on Other Metal Catalysts

Pandey and colleagues studied the kinetics of glycerol hydrogenolysis on a modified bi-functional CuNi/ $Al_2O_3$ catalysts [15]. The $Al_2O_3$-supported CuNi catalyst was prepared by the wetness impregnation method [53]. It was observed that the catalyst calcined at 400 °C gave the best performance of nearly 100% glycerol conversion and 89.5% selectivity to P at 220 °C and 0.75 MPa. The kinetic data were obtained in a packed bed down reactor under the conditions of a temperature range from 210–240 °C and a 0.75 MPa reaction pressure. An Eley–Rideal type kinetic modal was established in this work. In this model, the hydrogenolysis reaction of glycerol is thought to be carried out in three steps: glycerol adsorption, dehydration and direct hydrogenation with $H_2$ molecules. The following

equations were derived and obtained to account for the reaction rate of glycerol to acetol and acetol to P.

$$r_{G-A} = \frac{k_{G-A}K_GP_GC_{T\$}}{1 + K_GP_G + K_AP_A + P_PK_P}$$

$$r_{A-P} = \frac{k_{A-P}P_AP_{H_2}C_{T\$}}{K_A[1 + K_GP_G + K_AP_A + P_PK_P]}$$

The result showed that the activation energies of the dehydration of glycerol to acetol and hydrogenation of acetol to P were 55.1 kJ·mol$^{-1}$ and 50.9 kJ·mol$^{-1}$, respectively. The results were compared with the Langmuir–Hinshelwood model. It was proven that the CuNi/$\gamma$-Al$_2$O$_3$ catalyst can make the hydrolyzed glycerol reaction have a higher reaction rate at a lower temperature.

Liu and colleagues physically mixed Pd/ZrO$_2$ and ZnO for glycerol hydrogenolysis, confirming that direct use of physical mixtures leads to the in situ formation of active PdZn alloys on Pd surfaces [47]. The catalyst was tested in a 100 mL Teflon-lined stainless steel autoclave under the reaction conditions of 220 °C, 6 MPa H$_2$, 10 wt% glycerol in water with 10 wt% Pd loading. The result showed that ZnO plays an important role in increasing the reaction rate and selectivity, and the turnover rate and the selectivity to P were 90.2 mol$_{glycerol}$(mol$_{surface}$-$_{Pd}$·ks)$^{-1}$ and 94.1%, respectively.

The following model was proposed considering $\alpha$-C-H cleavage in 2,3-dihydroxypropa noxide to glyceraldehyde as the kinetically relevant step:

$$r_G = \frac{k_GC_GP_{H_2}{}^{-0.5}}{\left(1 + K_PC_G + K_{Alk}C_GP_{H_2}{}^{-0.5} + K_{H_2}{}^{-0.5}P_{H_2}{}^{0.5}\right)^2}$$

By fitting simulated data and experimental data, it was confirmed that the above equation can well describe the hydrogenolysis process of glycerol on PdZn surface. $\alpha$-C-H cleavage process forming the glyceraldehyde intermediate significantly affects the rate of hydrogenolysis. Therefore, the presence of Zn can make the fracture transition state of $\alpha$-C-H more stable, thus improving the conversion rate.

Xi and colleagues established a kinetic model of glycerol hydrogenolysis suitable for a trickle bed reactor [36]. The catalyst used in the experiment is PdReCo/C, with the bulk density in the trickle bed of approximately 700 kg·m$^{-3}$. The reaction conditions were a 40 wt% glycerol input concentration, 3.3–13.3 MPa H$_2$ and 180–203 °C, with 0.1–0.6 M NaOH added as a promoter. The reaction mechanism was proposed as follow: (Equation (1)) dehydrogenation of glycerol to an adsorbed glyceraldehyde analogue (GA·θ), (Equation (2)) rearrangement and dehydration of GA·θ to a second adsorbed intermediate (I·θ) analogous to pyruvaldehyde, and (Equation (3)) hydrogenation of the second intermediate to P.

$$G + \theta \leftrightarrow GA\cdot\theta + H_2 \tag{1}$$

$$GA\cdot\theta + OH^- \leftrightarrow I\cdot\theta + OH^- \tag{2}$$

$$I\cdot\theta + 2H_2 \rightarrow P + \theta \tag{3}$$

Therefore, the following equation can well represent the proposed mechanism.

$$r_G = \frac{k_GC_GC_{OH^-}C_{H_2}{}^2}{C_GC_{OH^-} + K_{H_2}C_{H_2}{}^3}$$

Compared with Cu-based catalysts, very limited fundamental understanding has been achieved on Ru and other metals for C-O cleavage of glycerol. However, it is clear that, consecutive hydrogenolysis reactions involving transformation of EG and P cannot be eliminated over Ru and other metals. This is because the adsorption of EG and P is strong over those metals. As a result, the overall selectivity for P is poor over those noble metals. Obviously, kinetic modeling only reflects the intrinsic rate of glycerol conversion

and the reaction rate of EG and P on the surface of catalysts. It cannot provide insights into plausible solution to reduce the significance of over hydrogenation reactions.

## 5. Discussion

(1) Current status of kinetic modeling. From a conventional reaction engineering point of view, both power law and LHHW model types can well reflect the intrinsic kinetics of glycerol conversion. The accuracy of various models can be acceptable for reaction engineering. However, one should note that most kinetic modeling was conducted under conditions with a low glycerol concentration (<20 wt%) except for the Cu catalyst. Therefore, prediction for kinetic trends cannot accurately reflect the reaction rates at higher glycerol concentrations.

Furthermore, existing problems with Cu- and Ru-based catalysts are not yet well characterized with molecular details. For example, deactivation of Cu catalysts caused by metal sintering has yet to be understood according to kinetic analysis. No relevant experimental studies have been dedicated to resolve this issue. As another example, the significant side reactions over Ru catalysts due to uncontrollable C-C cleavage leading to the formation of methane and methanol are yet to be well studied by chemical engineers. Although bimetallic catalysts can be potential options to improve selectivity, the acquisition of a fundamental understanding on an electronic (catalyst) and molecular level (computational calculation) has not even been attempted in this area. This is critical for reactor model development.

(2) Combinatory studies with cutting-edge technologies. Artificial intelligence can be used to assist decoupling of complicated reaction networks. In addition, computational fluid dynamics can be used to predict possible multiphase flow inside the pore of catalysts, which is important to predict chemo-selectivity within porous catalyst materials.

In addition, it is found that current kinetic modeling only depends on conventional characterization techniques. Various pieces of advanced characterization information on electronic and lattice levels are yet to be interpreted with experimental data.

Microkinetic studies have also been applied to analyze the rate of C-O and C-C cleavage of glycerol conversion. However, due to different metallic systems, those studies often generate contradicting results, which are difficult to reproduce for chemists and engineers [55–57].

## 6. Conclusions

Despite decade-long research efforts in the area of glycerol hydrogenolysis, the detailed pictures on how C-H and C-O bond activation occur on the surface of solid catalysts are still under debate. It can be seen from this critical review that it is generally believed that glycerol conversion is initiated by either a dehydrogenation or dehydration reaction over Cu-based catalysts. However, direct C-O cleavage over Ru-based catalysts does not seem to be a rate-limiting step compared with the dehydrogenation reaction. The latter one is more focused on co-adsorption of ethylene glycol rather than acetol as the intermediate. Compared with extensive studies on catalyst development, the kinetic modeling is very limited, which probably prevents engineers from gaining more insights into reactor modeling in existing catalytic systems. Future studies on kinetic modeling will be primarily focused on combinatory investigations into multiphase reactor modeling and durability improvement.

**Author Contributions:** Writing—original draft preparation, Y.L., J.L.; writing—review and editing, Z.X., X.Z.; supervision, C.L.; project administration, X.J.; funding acquisition, W.Y. All authors have read and agreed to the published version of the manuscript.

**Funding:** This research was funded by the National Natural Science Foundation (22078365, 22008262), the Natural Science Foundation of Shandong Province (ZR2020QB187), the Postdoctoral Research Funding of Shandong Province (201703016), the Qingdao Postdoctoral Research Funding (BY20170210), the "Fundamental Research Funds for the Central Universities" and "the Development Fund of State Key Laboratory of Heavy Oil Processing" (18CX02145A, 17CX02017A, 20CX02204A) and the new faculty start-up funding from the China University of Petroleum (YJ201601058).

**Conflicts of Interest:** The authors declare no conflict of interest.

## Nomenclature

| | |
|---|---|
| **r** | **Reaction Rate** |
| $K_i$ | equilibrium constant of component i |
| $K_H$ | adsorption equilibrium constant of hydrogen atom |
| $K_{ALK}$ | apparent equilibrium constant for the formation of bound 2,3-dihydroxypropanoxide from glycerol |
| ki | rate coefficient of i |
| $k_{P-PA}$ | reaction constant of hydrohydrolysis of 1,2-propanediol to 1-propanol |
| $P_H$ | hydrogen atom pressure |
| G | glycerol |
| $H_2$ | hydrogen |
| P | 1,2-propanediol |
| EG | ethylene glycol |
| A | acetol |
| W | water |
| iG | ethylene glycol and 1,2-propanediol |
| $C_{T\$}$ | total concentration of active sites |
| $C_i$ | concentration of component i |
| $C_{OH^-}$ | base concentration |
| $\theta_i^*$ | the surface coverages of surface species i |
| $\theta^*$ | the fractional coverage of free sites |
| $S_{Ig}$ | the respective selectivity factors |
| H | Henry's constant |
| $r_G$ | total reaction rate of glycerol consumption |
| $r_{G-P}$ | reaction rate of hydrohydrolysis of glycerol to 1,2-propanediol |
| $r_{G-EG}$ | reaction rate of hydrohydrolysis of glycerol to ethylene glycol |
| $r_{G-A}$ | reaction rate of glycerol to acetol |
| $r_{A-P}$ | reaction rate of acetol to 1,2-propanediol |
| $r_{iG}$ | degradation rate of EG and P |

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
