# Peer review of "Kinetic Modeling of Glycerol Hydrogenolysis: A Short Review"

_catalysts, doi:10.3390/catal13010023_

Round 1

Reviewer 1 Report

This manuscript reviews the reports for kinetic analysis of glycerol hydrogenolysis to 1,2-propanediol. Generally, this manuscript well summarized the literature and is informative. I suggest the following points for improvement of this manuscript.

1. This reaction is typically operated in the presence of water, and the most catalysts contain non-noble transition metals. The reduction degree of such component in the catalysts may depend on the reaction conditions, especially on hydrogen pressure. The difference of reduction degree affects the catalytic activity, complicating the kinetics on the hydrogen pressure. This point should be mentioned.

2. The Introduction section should be improved. It should be emphasized that the production of 1,2-propanediol (P) from glycerol has been already commercialized. The use of P as antifreeze should be mentioned in the main text. For 1,3-PDO, the reference [16] is too old. The production of 1,3-PDO has been intensively investigated in these years. Recent review papers should be cited.
For ethylene glycol (EG), the use of bioethylene from ethanol should be mentioned for bio-EG.

3. Mono-alcohols are not the final products of the hydrogenolysis. Some systems produce alkanes (propane and ethane). The term "final" is not appropriate.

4. In EG production from glycerol, the rest C1 fragment is usually obtained as CH4 which is an undesirable product. Figure 2 should describe CH4 after methanol.

5. The usage of technical terms should be closely checked again. Hydrogenation and hydrogenolysis are different reactions and should be used apart. "power law model" is sometimes miswritten as "power low model".

6. These papers can be included.
Appl. Catal. A 2019, 576, 47 (10.1016/j.apcata.2019.03.001)
React. Chem. Eng. 2019, 4, 595 (10.1039/C8RE00138C)
Biomass 2022, 2, 27 (10.3390/biomass2010003)
Environ. Technol. Innovation 2022, 27, 102367 (10.1016/j.eti.2022.102367)

Reviewer 2 Report

The article is very nice and I do not have any problems to publish it. From reactions to kinetic equations, it is very clear. Please just remove citations from the Conclusions.
